# Medical students' crisis-induced stress and the association with social support

Vera M. A. Broks[1], Karen M. Stegers-Jager[1], Jeroen van der Waal[2], Walter W. van den Broek[1], Andrea M. Woltman[1]*

1 Institute of Medical Education Research Rotterdam, Erasmus MC University Medical Centre, Rotterdam, The Netherlands, 2 Department of Public Administration and Sociology, Erasmus University Rotterdam, Rotterdam, The Netherlands

* a.woltman@erasmusmc.nl

## Abstract

### Background

Medical schools are challenged to guard student wellbeing given the potential negative impact of the COVID-19 outbreak combined with an already high prevalence of mental distress. Although social support is generally associated with less crisis-induced stress, it is unknown whether this applies to medical students during the COVID-19 outbreak.

### Objectives

The impact of the COVID-19 outbreak on perceived stress of medical students was assessed by comparing their perceived stress levels during the outbreak to both their own baseline and the previous cohort's pre-COVID-19 stress levels. Then, the association between social support and stress during the COVID-19 outbreak was assessed.

### Methods

Dutch Year-1 medical students of cohort 2019 ($n = 99$) completed the 14-item Perceived Stress Scale (PSS-14) at two time points: baseline (pre-COVID-19) and final measurement (COVID-19). Social support—emotional-informational support and club membership—was assessed during the final measurement. PSS and social support scores were compared to similar measurements of cohort 2018 ($n = 196$). Students' baseline stress levels, gender, and study performance were controlled for when comparing final stress levels.

### Results

In cohort 2018 (pre-COVID-19), students' perceived stress levels did not differ significantly between the baseline and final measurements. Additionally, baseline stress levels of the two cohorts (2018 and 2019) were not found to be significantly different. Cohort 2019's final stress levels (COVID-19) were significantly higher compared to their baseline stress levels (paired t-test: t = 6.07, $p < .001$) and cohort 2018's final stress levels (linear regression: B = 4.186, $p < .001$). Only during the COVID-19 outbreak higher social support

**Data Availability Statement:** The database contains levels of perceived stress linked to students personal characteristics. Due to the

sensitivity of the data we chose to not make these data publicly available. Anonymized data is available upon request. Data requests may be sent to the institute of Medical Education Research Rotterdam (iMERR). Requests can be sent to associate professor Sílvia Mamede (iMERR): s. mamede@erasmusmc.nl. iMERR can also be contacted via the website: www.imerr.nl, under the tab "Contact".

**Funding:** The author(s) received no specific funding for this work.

**Competing interests:** The authors have declared that no competing interests exist.

levels—i.e., emotional-informational support (B = -0.75, $p < .001$) and club membership (B = -3.68, $p < .01$)—were associated with lower stress levels.

## Conclusions

During the COVID-19 outbreak, medical students' perceived stress levels were higher—especially for students with lower social support levels. Our results suggest that medical schools should optimize social support to minimize crisis-induced stress.

## Introduction

The prevalence of mental distress, such as symptoms of anxiety, depression, or burnout, in medical students is high compared to their age-matched peers [1–3]. Approximately a quarter to one-third of medical students show symptoms of depression [4, 5], and roughly 40% experience burnout symptoms [6]. These mental problems can be caused by stress [7]. A recent stressor is the COVID-19 outbreak. The potential negative impact of the outbreak on mental wellbeing combined with the already high prevalence of mental problems in medical students, exacerbates the challenge medical schools face in guarding their students' wellbeing [8]. Research regarding factors related to higher stress levels during a crisis or in other situations in which stressors increase, will enable medical schools to limit the negative impact of such crises on student wellbeing. Social support is possibly one of the factors associated with crisis-induced stress [9–11]. Therefore, the present study had two objectives. The first objective was to investigate how the COVID-19 outbreak has impacted the perceived stress of medical students. The second objective was to investigate the association between social support and stress during the COVID-19 outbreak for medical students.

The COVID-19 outbreak has disrupted everyday life, which has negatively impacted the mental wellbeing of the general population [12–14]. Compared to the general population, students' mental health has been more affected by the outbreak [15]. A possible explanation is that student life and its social aspects have been affected by measures such as social distancing, lockdown, quarantine [16], and the transition to online education [17]. However, for medical students, studies showed mixed results regarding the impact of the COVID-19 outbreak on wellbeing [18, 19]. A systematic review reported that anxiety levels in medical students did not increase during the outbreak [18]. Other studies have reported higher levels of burnout symptoms and stress for medical students during the outbreak [19], especially in female students [20, 21]. However, these studies often consisted of self-reported increased stress levels without a baseline measurement [20, 21], or did not correct for probable changes in stress levels throughout the academic year regardless of the outbreak [19]. Finally, although a negative relationship between study performance and stress is known for medical students [22], the role of academic performance was not taken into account in previous studies. In summary, previous studies concerning the impact of the COVID-19 outbreak often lack controls for baseline measurements and student performance.

According to the stress-buffering model, the negative impact of a stressful event on wellbeing is stronger for individuals with less social support [10, 11]. Social support helps a person to appraise certain events as less stressful. In addition, social support alleviates the impact of the appraised stress (e.g., by offering a solution or reducing the importance). Strong and weak ties can be distinguished in social support systems, such as intimate ties with friends or family and non-intimate ties with acquaintances, respectively [23]. Intimate

ties (i.e., strong ties) can offer emotional-informational support, composed of an emotional and informational component. The emotional component consists of care, love, and empathy (e.g., providing a listening ear) [24]. The informational component consists of guidance to offer a solution for a problem (e.g., providing advice) [24]. Non-intimate ties (i.e., weak ties), such as ties with fellow members of a sports- or hobby club, are less personal [23, 25], but can offer social companionship involving time spent with others in activities of recreational nature [24]. According to the literature, both strong and weak ties can positively affect wellbeing and can buffer stress [25–27]. Social support therefore refers to people whom someone can turn to in times of crisis—such as family, friends, or fellow club members. A literature review showed that after a hurricane, tsunami, or terrorist attack, social support was associated with higher resilience [28]. During the COVID-19 outbreak, literature concerning the general population indicated that social support was associated with a decreased sense of loneliness [29], and increased resilience [30]. For adolescents and college students, social support during times of COVID-19 was associated with fewer symptoms of depression and anxiety [31], and stress [32]. For medical students, social support is positively linked to mental wellbeing [33–39]. However, to the best of our knowledge, it is not yet known whether social support is associated with crisis-induced stress among medical students, especially in a crisis that strains social contacts through all kinds of social distancing measures.

The COVID-19 outbreak presented a unique opportunity to assess medical students' crisis-induced stress on a large scale. Stress is linked to both psychological and physical long-term wellbeing and can be caused by environmental factors such as the COVID-19 outbreak [7, 40–42]. However, existing studies cannot provide solid evidence of altered stress levels during the COVID-19 outbreak due to the lack of required baseline measurements and controls for study performance. Even though medical schools are unable to resolve the present COVID-19 crisis or any future crisis, they might be able to limit the negative consequences on student wellbeing. To get more insights into how medical schools can provide the right support to their students in times of crisis, we aimed to answer the following research questions:

1. What was the impact of the COVID-19 outbreak on medical students' perceived stress compared to both their own baseline stress level and compared to the stress level of the previous cohort, while controlling for gender and study performance?

2. Did social support, broken down into emotional-informational support and club membership, moderate the effect of the COVID-19 outbreak on the perceived stress of medical students?

## Methods

### Context

The present study was conducted with Year-1 Bachelor students of Erasmus MC Medical School in the Netherlands. Dutch medical schools consist of a 3-year Bachelor's- and 3-year Master's program. At Erasmus MC Medical School, the Bachelor curriculum is composed of preclinical training in thematic blocks and competence-based learning lines. Each year, 60 credits under the European Credit Transfer System (ECTS) can be obtained, resulting in 180 credits for the complete Bachelor program. Grades are based on a 10-point scale from 1 to 10 (maximum) where 5.5 is the minimum to pass. In March 2020, the COVID-19 outbreak started to impact everyday life in the Netherlands. For the Bachelor students of Erasmus MC

Medical School, this entailed that all on-site classes were dismissed and online education became the new standard.

## Participants and procedure

All Year-1 Bachelor students from Erasmus MC Medical School, who enrolled in cohorts 2018 (409 students) and 2019 (408 students) were invited to participate by completing a questionnaire regarding perceived stress in December/January (baseline measurement—online questionnaire) and May (final measurement—cohort 2018: a combination of paper and online, cohort 2019: online) of their first academic year. During the final measurement, social support was measured in addition to perceived stress. The sample in the present study consisted of students who completed both questionnaires. Only the final measurement of cohort 2019 took place during the COVID-19 outbreak (see Fig 1A). Students provided written informed consent for the data collected by questionnaires and for linking questionnaire results to relevant data from the student administration. The university student administration provided data regarding gender and study performance up to the final measurement. To link data from different sources, students' unique identification numbers were used. After linking the data, these student identification numbers were removed from the database. The student administration also provided aggregated data from the complete cohorts, including students that did not complete the questionnaires. Since these data were only reported on an aggregated level, no individual consent was required. The study was carried out in accordance with the Declaration of Helsinki and was deemed exempt from review after evaluation by the Medical Ethics Committee of Erasmus MC Rotterdam (MEC-2019-0448).

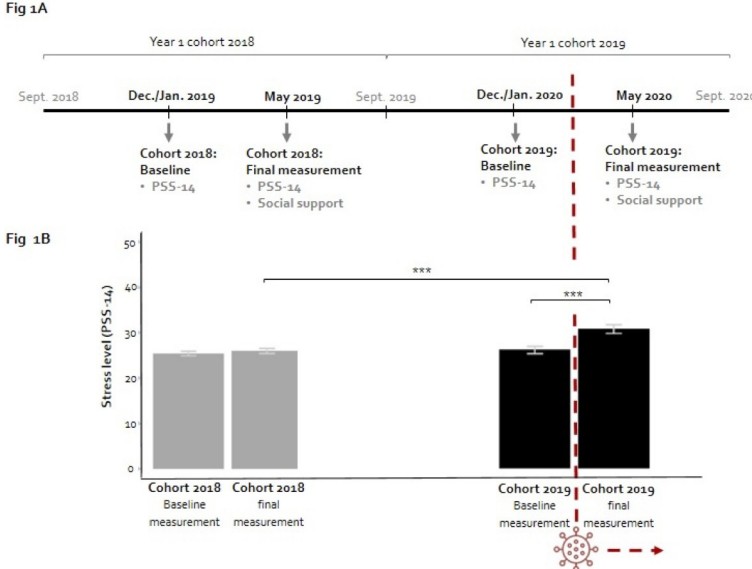

**Fig 1. Timeline of data collection and mean perceived stress level for each measurement. A**. Timeline of data collection. The COVID-19 pandemic began to impact everyday life and medical school during Year-1 of cohort 2019 (March 2020). **B**. Mean perceived stress levels including error bars. Mean (M) stress levels and Standard Errors (SE) are shown for the baseline measurement (M = 25.38, SE = 0.51) and the final measurement (M = 26.00, SE = 0.54) for cohort 2018 and baseline measurement (M = 26.05, SE = 0.79) and the final measurement (M = 30.65, SE = 0.99) for cohort 2019. Differences were assessed within a cohort with a paired t-test. The difference between the final measurement of cohort 2018 and cohort 2019 was assessed in a linear regression model while controlling for baseline measurement, gender, and study performance (Table 2–Model 1). Significant differences between measurements are shown: *** $p < .001$.

## Measurements

**Student characteristics.** Student characteristics that were taken into account were gender and study performance. Gender was categorized as male or female. Student performance was operationalized by measuring whether students obtained all possible credits up until the final measurement in May of Year-1 (yes/no). The maximum number of obtainable credits up until the final measurement was equal to 38 credits for cohort 2018 and equal to 25 credits for cohort 2019. This difference in credits was due to the fact that for cohort 2019, some exams were postponed due to the COVID-19 outbreak along with the transition to online education.

**Perceived stress (PSS-14).** Perceived stress was measured with a Dutch version of the 14-item perceived stress scale (PSS-14) [43, 44]. As recently described, this questionnaire is appropriate to measure stress responses [42] and measures general distress and someone's ability to cope with this stress. The questionnaire focuses on students' feelings during the last month, for example, "In the last month, how often have you been upset because of something that happened unexpectedly?". The items are scored on a 5-point Likert scale ranging from 0 (never) to 4 (very often). The maximum score on the PSS-14 is 56.

**Social support.** Social support was assessed with questions regarding emotional-informational support and club membership to cover both stronger and weaker social ties that are relevant to students in the Netherlands [45]. To measure emotional-informational support, four items of the MOS social support scale were selected from the subscale emotional-informational support [24] (S1 Table). For the selection of the items, the following was taken into account: an equal distribution of both emotional and informational items, item-scale correlations, and applicability to our sample. The selected items focus on the availability of different forms of social support, such as having someone to listen to you, offer you advice, or share your thoughts with. The items were scored on a 5-point Likert scale ranging from 0 (never) to 4 (always). The maximum score on emotional-informational support was 16. Alpha reliability measures were computed to assess internal consistency for this adjusted scale (S1 Table). The second operationalization of social support was club membership. Club membership was measured with the closed yes/no question "Are you member of a hobby club, sports club or a leisure club?".

## Statistical analysis

Analyses were only performed on results of students who fully completed the questionnaires of both the baseline and final measurement during Year-1. First, the samples of the two included cohorts were compared on response rates, gender, study performance, social support, and stress levels. T-tests and Wilcoxon rank-sum tests were used for the comparison of normally and non-normally distributed continuous variables respectively. Dichotomous variables were compared between cohorts with chi-square tests. Changes in stress levels during the COVID-19 outbreak were assessed using multiple methods. Firstly, paired t-tests were performed in both cohorts to assess whether stress levels within the cohorts changed between their baseline and final measurement. Secondly, final stress levels of cohort 2018 (pre-COVID-19) and cohort 2019 (COVID-19) were compared in a linear regression model while controlling for baseline stress level, gender, and study performance. The effect of social support on perceived stress was assessed by adding emotional-informational support and club membership to the linear regression model. Finally, the interaction terms of social support and cohort were added to the linear regression model to investigate whether the relation between social support and perceived stress was different during the COVID-19 outbreak. All continuous variables in the regression model were centred for interpretation purposes. The regression model met all assumptions, and Cook's distance revealed no observations that had a large impact on the

regression equation [46]. Therefore, regression analyses were performed without modifications. Finally, to illustrate the results of the linear regression analysis, four subgroups were formed based on the presence of club membership (yes/no) and emotional-informational support (high/low). Emotional-informational support was considered low when students' scores were in the 25th percentile of the complete sample. This resulted in a reasonable cut-off to select students that scored lower than the median while still maintaining an acceptable group size.

## Results

### Cohort characteristics

The response rate was lower in cohort 2019 compared to cohort 2018 (24% vs. 48%, Table 1). Proportions of female students and of students who acquired all obtainable credits until the final measurement was comparable between cohorts (see Table 1). Though emotional-informational support scores remained the same (Table 1), a chi-square test showed that the percentage of students that were members of a club was significantly lower in cohort 2019: 56% in cohort 2019 compared to 68% in cohort 2018 ($\chi2$ = 4.150, df = 1, $p$<0.05). The mean baseline stress level was found to be comparable between cohorts, but the mean stress level during the final measurement was significantly higher for cohort 2019 than for cohort 2018 ($t$ = 4.134, df = 158.69, p < .001).

### Perceived stress levels

A paired t-test demonstrated that the perceived stress levels of cohort 2019 significantly increased from 26.05 to 30.65 (t = 6.07, df = 98, $p$ < .001), whereas the stress levels of cohort 2018 did not significantly differ between the baseline and final measurement (Fig 1B). This indicates that the perceived stress levels of students significantly increased during the COVID-19 outbreak.

To assess the effect of the COVID-19 outbreak, we not only compared stress levels *within* cohorts, we also compared the stress levels during the final measurement *between* two cohorts of medical students. For the control variables, a higher baseline stress level and not having obtained all obtainable credits were related to higher stress levels during the final measurement (Table 2–Model 1). When controlling for baseline stress level, gender, and study performance, a significantly higher perceived stress level was visible for cohort 2019 (COVID-19) compared to cohort 2018 (Pre-COVID-19; $B_{cohort}$ = 4.186, 95%-CI: 2.608–5.764, $p$ < .001, Table 2–

**Table 1. Descriptive statistics of variables included in the analyses.**

| | Cohort 2018 | | Cohort 2019 | | Comparison of cohorts (samples) |
|---|---|---|---|---|---|
| | Total | Sample | Total | Sample | |
| **Number of students (% of total)** | 409 (100) | 196 (48) | 408 (100) | 99 (24) | $\chi2$ = 48.529, df = 1, p < .001 |
| **Female students: %** | 74 | 79 | 68 | 74 | $\chi2$ = 0.787, df = 1, p = 0.375 |
| **Students with all credits obtained until final measurement: %** | 66 | 71 | 65 | 67 | $\chi2$ = 0.378, df = 1, p = 0.539 |
| **Social support** | | | | | |
| **Median emotional-informational support: median/total (range)[a]** | - | 14/16 (3–16) | - | 14/16 (2–16) | W = 10250, p = 0.419 |
| **Club member: % yes** | - | 68 | - | 56 | $\chi2$ = 4.150, df = 1, p < .05 |
| **Stress level (PSS-14)** | | | | | |
| **Mean (sd) baseline measurement** | - | 25.38 (7.17) | - | 26.05 (7.86) | t = -0.710, df = 181.62, p = 0.479 |
| **Mean (sd) final measurement** | - | 26.00 (7.57) | - | 30.65 (9.80) | t = 4.134, df = 158.69, p < .001 |

[a] Data are not normally distributed, non-parametric Wilcoxon rank-sum test was used to compare cohorts.

**Table 2. Results linear regression model with outcome variable stress level (PSS-14) in the final measurement.**

| | Model 1 | | Model 2 | | Model 3 | |
|---|---|---|---|---|---|---|
| | B [95% CI] | sig. | B [95% CI] | sig. | B [95% CI] | sig. |
| *Intercept* | 27.132 [24.985–29.280] | - | 27.670 [25.323–30.017] | - | 26.977 [24.466–29.487] | - |
| **PSS—baseline**[a] | 0.616 [0.511–0.721] | *** | 0.567 [0.461–0.672] | *** | 0.579 [0.474–0.684] | *** |
| **Female** (ref: male) | 1.766 [-0.061–3.594] | . | 2.231 [0.433–4.030] | * | 2.163 [0.384–3.943] | * |
| **All credits obtained** (ref: not all credits) | -3.372 [-5.025–-1.718] | *** | -3.637 [-5.260–-2.016] | *** | -3.427 [-5.041–-1.813] | *** |
| **Cohort 2019—COVID-19** (ref: 2018) | 4.186 [2.608–5.764] | *** | 3.894 [2.341–5.446] | *** | 5.220 [2.694–7.747] | *** |
| **Emotional-informational (E-I) support**[b] | | | -0.417 [-0.659–-0.177] | *** | -0.230 [-0.532–0.0714] | |
| **Club member: yes** (ref: no) | | | -1.636 [-3.197–-0.077] | * | -0.501 [-2.429–1.428] | |
| **Cohort (2019) * E-I support** | | | | | -0.516 [-1.004–-0.027] | * |
| **Cohort (2019) * Club member (yes)** | | | | | -3.180 [-6.333–-0.028] | * |
| **Adjusted R²** | 0.436 | | 0.462 | | 0.474 | |

Note: Dependent variable: stress level during the final measurement: score on the PSS-14 (min = 0, max = 56). "B" refers to the unstandardized regression coefficient together with the 95% confidence interval (95% CI). For each regression coefficient, the table shows whether it significantly deviates from 0 ($H_0$: B = 0, $H_A$: B≠0) in the column "sig." using the following annotation:

*** $p < .001$,

**$p < .01$,

*$p < .05$,

˙$p < .1$.

The continuous variables are centred:

[a] PSS = baseline is centred on the mean = 25.61;

[b] Emotional-informational support is centred on the median = 14.

Model 1 and Fig 1B). Compared to cohort 2018, the perceived stress levels for students of cohort 2019—during the COVID-19 outbreak—were on average approximately 4 units higher on the Perceived Stress Scale (ranging from 0 to 56).

## Social support

When taking into account social support, perceived stress levels were still significantly higher during the COVID-19 outbreak ($B_{cohort}$ = 3.894, 95%-CI: 2.341–5.446, $p < .001$, Table 2–Model 2). In addition, it was found that students' perceived stress levels were lower when a student experienced more emotional-informational support($B_{E-I\ support}$ = -0.417, 95%-CI: -0.659–-0.177, $p < .001$, Table 2–Model 2). Also, students who were club members had significant lower stress levels compared to their fellow students who were not ($B_{club\ member}$ = -1.636, 95%-CI: -3.197–-0.077, $p < .05$, Table 2–Model 2).

Finally, the moderating effects of emotional-informational support and club membership with cohort were included. Both the effects of emotional-informational support ($B_{E-I\ support*cohort}$ = -0.516, 95%-CI: -1.004–-0.027, $p < .05$) and club membership ($B_{club\ member*cohort}$ = -3.180, 95%-CI: -6.333–-0.028, $p < .05$; Table 2–Model 3) were found to be significantly different for cohort 2019 (COVID-19) compared to cohort 2018 (pre-COVID-19). Only for cohort 2019, stress levels were significantly lower for students with higher levels of emotional-informational support (Fig 2A). Regarding club membership, only cohort 2019 demonstrated a significant difference in stress level between students who were club member or not (Fig 2B). These results suggest that both emotional-informational support and club membership were associated with lower stress levels for students in cohort 2019. Fig 3 illustrates that cohort 2019 students with only one of the two types of social

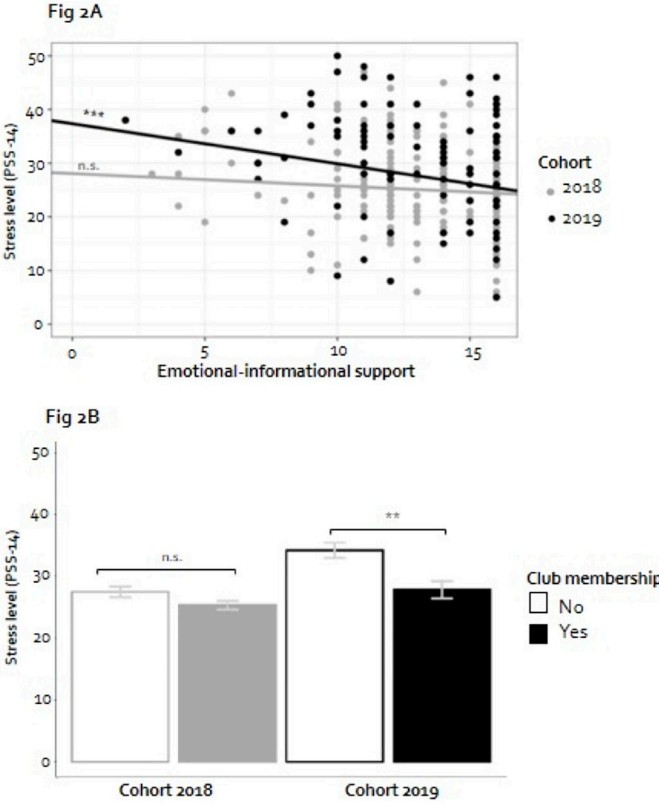

**Fig 2. Social support and perceived stress level during the final measurement per cohort. A**. Emotional-informational support. Each dot represents an observation and demonstrates the perceived stress level and the level of emotional-informational support of an individual student. The slope of the regression line represents the effect of emotional-informational support on stress levels. The slope of the regression line in grey for cohort 2018 is equal to -0.23 (non-significant, Table 2–Model 3). The slope of the regression line in black for cohort 2019 is -0.75 (p < .001; p-value is based on a post-hoc analysis), which equals the sum of the regression coefficients for emotional-informational support and emotional-informational support*cohort 2019 = -0.23–0.52 = -0.75 (Table 2–Model 3). **B**. Club membership. Observed mean perceived stress levels including error bars (M±SE) for students without and with club membership. In the linear regression model, for cohort 2018, no significant effect is present for club membership (coefficient club member = -0.50, non-significant, Table 2–Model 3). For cohort 2019, the linear regression model shows a difference of -3.68 (p < .01; p-value is based on a post-hoc analysis), which is the sum of the regression coefficients for club member and club member*cohort 2019: -0.50–3.18 = -3.68 (Table 2–Model 3).

support–club membership *or* high emotional-informational support—showed significantly higher stress levels compared to students with both types of social support. For cohort 2018 students, differences in stress levels based on social support were not present.

## Discussion

The present study demonstrates that medical students' perceived stress levels were significantly higher during the COVID-19 outbreak—compared to their own pre-COVID-19 baseline stress levels as well as to the stress levels of students in the previous pre-COVID-19 cohort. During the outbreak, students who experienced less emotional-informational support or were not a member of a club showed higher perceived stress levels compared to their fellow students with more emotional-informational support or who were a member of a club. The present results indicate that during times of crisis, social support is associated with less perceived stress in medical students.

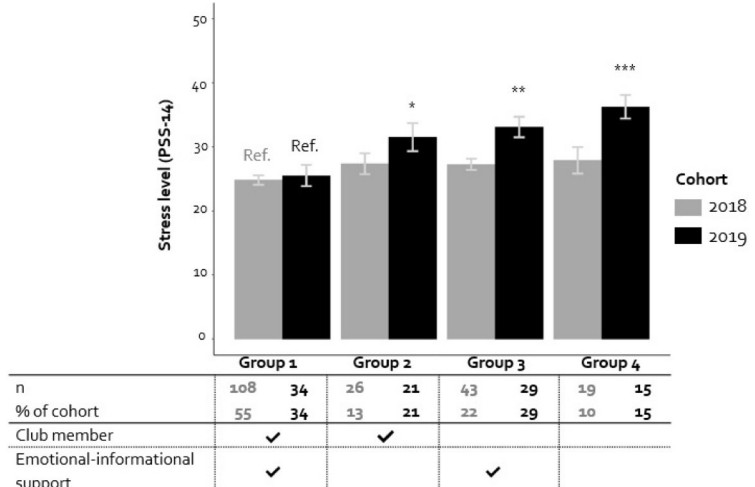

**Fig 3. Perceived stress level of groups during the final measurement based on club membership and emotional-informational support.** Mean perceived stress level for groups based on club membership and emotional-informational support for cohort 2018 and cohort 2019 including error bars. Emotional-informational support is unchecked if the score is ≤25th percentile of the complete sample (score ≤11). For each cohort separately, the stress levels of groups 2 to 4 were compared to the stress level of group 1 in a post-hoc linear regression analysis (reference group = Ref.). For cohort 2018, groups 2 to 4 did not differ significantly from group 1 regarding stress levels. For cohort 2019, groups 2 to 4 did differ significantly from group 1 regarding stress level: $^*p < .05$, $^{**}p < .01$, $^{***}p < .001$, respectively.

Our finding that medical students' stress levels during the COVID-19 outbreak were higher compared to pre-COVID-19 stress levels, corresponds with previous studies reporting higher stress levels during the outbreak [19–21]. More specifically, we found that students' stress levels increased compared to their own baseline stress levels as well as compared to the stress levels of a previous cohort, controlling for gender and study performance. Also, we took into account potential fluctuations during the academic year regardless of the pandemic, as reported in a previous study [47]. The mean stress level reported in the present study during the outbreak was higher than the mean stress level of Dutch medical students after increased performance standards [44]. Still, these reported stress levels did not exceed the stress levels of medical students in the US after the implementation of a new curriculum [48]. The reason why stress levels were elevated during the COVID-19 outbreak goes beyond the scope of the present study, but one might speculate about a mix of academic uncertainty [8, 17], online education [20], blurred study-home boundaries, and social isolation [49]. Students' stress levels are subjected to a variety of factors and are therefore context-specific. This context-specificity of students' stress levels illustrates the importance of baseline measurements to make valid comparisons. Such baseline measurements could also serve future research on the long-term effects of the COVID-19 outbreak.

In line with the stress-buffering model [10, 11], we found that only during the COVID-19 outbreak, in which social contacts were strained by social distancing measures, higher levels of social support were associated with lower stress levels for medical students. This mechanism has also been confirmed by a recent study in which medical students reported that their own most effective strategies to lower the negative impact of stress included de-stress through friends and family [50]. In line, in times of the COVID-19 outbreak, social support was found to be positively associated with wellbeing in adolescents [31], college students [32], and the general population [29]. The present study adds to these findings in two ways. Firstly, we

provide evidence that this also applies to medical students; a group that already experiences higher levels of mental distress compared to their peers [1–3]. Secondly, our study shows that the association between social support and stress level also applies in times of crisis where social support is strained due to social distancing measurements [12, 13]. Although students did not report lower levels of emotional-informational support during the COVID-19 outbreak, the number of students who reported being club members was lower during the COVID-19 outbreak than before (56% vs. 68%). This lower percentage is possibly the consequence of measures taken to limit the spread of the coronavirus, making it impossible for clubs to gather with their members. Emotional-informational support was not lower during the COVID-19 outbreak. A potential explanation is that students moved back home to their parent's house where emotional-informational support was still available since close family is a source of social support for Dutch students [45]. Perhaps, the *amount* of emotional-informational support did not change but the *source* of this support did: from school-based relationships to relationships with close family. This could be further explored in future research.

The present study illustrates how two forms of social support—emotional-informational support and club membership—are complementary to each other in relation to reported stress levels during the COVID-19 outbreak. An explanation for this finding can be found in a theoretical model for mechanisms linking social support to health. This model distinguishes primary and secondary social resources, described as *intimates* and *knowledgeable others* respectively [25]. According to this model, these two social sources independently attribute to buffering the impact of stressors. Whereas *intimates* buffer the stress by companionate presence, offering care and instrumental assistance, *knowledgeable others* buffer stress by enabling ventilation and role modelling. Emotional-informational support can be considered a primary source (i.e., stronger ties: intimates) whereas club membership can be considered a secondary source of social support (i.e., weaker ties: knowledgeable others). This possibly explains the added value of combining both emotional-informational and club membership. Yet, it should be noted that club membership reflects more than the level of social support, as previous research also showed its link to socioeconomic status. For example, it was shown that adolescent girls with lower socioeconomic status are less likely to participate in a sports club [51]. Therefore, the lack of club membership may also reflect a lower socioeconomic status, which in turn negatively affects self-reported health [52]. Indeed, during the COVID-19 outbreak, university workers living in smaller homes reported higher levels of anxiety- depression- and stress symptoms [53]. Also, for college students during the COVID-19 outbreak, economic disadvantage was associated with higher stress levels [54]. Perhaps, the higher levels of stress found in the present study for students without club membership are also partially explained by their socioeconomic status, but more research is needed.

A strength of the current study is that two measurements within two cohorts were included, enabling controls for baseline measurements and previously reported fluctuations throughout the academic year [47]. Moreover, gender and study performance was controlled for, which are known to be correlated with stress levels [20, 22, 44]. Even though study performance was taken into account, the meaning of having obtained all possible credits up until the final measurement differed between cohorts. Due to the outbreak, exams were postponed which resulted in a higher number of exams that still had to be completed in the last part of the academic year. We cannot rule out any stress caused by these postponed exams during the outbreak as described in a previous study [55]. A limitation of the present study is the lower response rate for cohort 2019 compared to cohort 2018. This was possibly due to the COVID-19 outbreak, which resulted in a fully online data collection instead of a combination of data collection in class and online. For both cohorts, students who only completed the baseline measurement did not have significantly different stress levels compared to students who

completed both measurements (cohort 2018: t = -0.220, df = 172.87, p = .826; cohort 2019: t = 1.229, df = 198.15, p = .221). Despite these results and the fact that we controlled for student characteristics in the analysis, a selection bias may still be present due to the lower response rate. Perhaps, the highly stressed students were less likely to complete the questionnaires because they did not want to be bothered with them. On the other hand, it is also possible that the students who were highly stressed were more likely to complete the questionnaire because of the relevance of the topic to them, as opposed to students who were feeling less stressed. This may have led to an underestimation or overestimation of the effect. Another limitation of the present study is that we investigated the levels of perceived stress, but not the cause of per- ceived stress. Higher stress levels during the COVID-19 outbreak could have been caused by either direct effects of the pandemic itself, or indirect effects due to changes in the student's environment (e.g., living situation, less contact with the faculty). Additionally, the present study focused on Year-1 students, and this appears to be a relevant group since students in the early stages of medical and dental school seem susceptible to the negative impact of the COVID-19 outbreak [19, 56]. Whether the results described in the present study will be similar for students in advanced stages of medical school needs to be further investigated. This also applies to the generalizability of the results to students from other schools. The relation between social support and students' perceived stress may be different between schools as the prevalence of mental distress is higher for medical students compared to their age-matched peers [1, 2].

Even though medical schools are not able to change the current COVID-19 crisis or any future crisis, they can help students get through it. Our findings suggest that in times of crisis medical students' wellbeing can benefit from social support. With the COVID-19 pandemic still not being over and potential new lockdowns being possible, medical schools could play a more active role by further exploring the (digital) options to provide different kinds of social support to their students. The first kind of social support entails companionate care and instrumental assistance, which can be provided through one-on-one (online) mentoring with a faculty member [57], or a peer [58, 59]. The second kind of social support has more to do with enabling ventilation with peers and role modelling, some sort of social embeddedness. Medical schools can achieve this by for example creating online communication platforms [60], or by stimulating peer relationships by promoting cooperation amongst students in the medical school program [61, 62]. All in all, when implementing (online) education, medical schools should not only focus on qualifications but also on the social functions of education.

To conclude, the present study provides solid evidence of higher levels of stress during the COVID-19 outbreak in medical students, especially among those with less social support. The findings of the present study are in accordance with an existing model describing the buffering effect of social support on crisis-induced stress and therefore go beyond the current COVID-19 pandemic. Medical schools can optimize social support for their students by offering social support on different levels to minimize the negative impact of future global, national or indi- vidual crises.

## Supporting information

**S1 Table. Measurement emotional-informational support.** *Note*: Questions and scoring emotional-informational support in English, with in italic-grey the Dutch version used for the data-collection. Minimum score: 0, maximum score: 16. Alpha reliability cohort 2018 = 0.895. Alpha reliability cohort 2019 = 0.907.
(PDF)

## Acknowledgments

We thank Daphne Pol, who helped with data collection and data entry of cohort 2018 for our study. We also wish to thank David van Klaveren for helping us with our statistical analyses.

## Author Contributions

**Conceptualization:** Vera M. A. Broks, Karen M. Stegers-Jager, Jeroen van der Waal, Walter W. van den Broek, Andrea M. Woltman.

**Data curation:** Vera M. A. Broks.

**Formal analysis:** Vera M. A. Broks.

**Investigation:** Vera M. A. Broks, Karen M. Stegers-Jager, Andrea M. Woltman.

**Methodology:** Vera M. A. Broks, Karen M. Stegers-Jager, Jeroen van der Waal, Andrea M. Woltman.

**Supervision:** Karen M. Stegers-Jager, Walter W. van den Broek, Andrea M. Woltman.

**Visualization:** Vera M. A. Broks.

**Writing – original draft:** Vera M. A. Broks.

**Writing – review & editing:** Vera M. A. Broks, Karen M. Stegers-Jager, Jeroen van der Waal, Walter W. van den Broek, Andrea M. Woltman.

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
