## [Decision Letter · Decision Letter 0]

18 Apr 2022

PONE-D-21-39400Medical students’ crisis-induced stress and the association with social supportPLOS ONE

Dear Dr. Broks,

Thank you for submitting your manuscript to PLOS ONE. After careful consideration, we feel that it has merit but does not fully meet PLOS ONE’s publication criteria as it currently stands. Therefore, we invite you to submit a revised version of the manuscript that addresses the points raised during the review process.

You can see the reviewers' comments at the end of this email. While the reviewer 2 has indicated some stylistic errors, reviewer 1 has pointed out issues with variables in Introduction as well as using different statistical tests in in the methods and analysis section.

We look forward to receiving your revised manuscript.

Kind regards,

Muhammad A. Z. Mughal, PhD

Academic Editor

PLOS ONE

Journal Requirements:

Reviewers' comments:

Reviewer's Responses to Questions

**Comments to the Author**

1. Is the manuscript technically sound, and do the data support the conclusions?

Reviewer #1: Yes

Reviewer #2: Yes

2. Has the statistical analysis been performed appropriately and rigorously? 

Reviewer #1: I Don't Know

Reviewer #2: Yes

3. Have the authors made all data underlying the findings in their manuscript fully available?

Reviewer #1: No

Reviewer #2: Yes

4. Is the manuscript presented in an intelligible fashion and written in standard English?

Reviewer #1: Yes

Reviewer #2: Yes

5. Review Comments to the Author

Reviewer #1: Manuscript #PONE-D-21-39400 examined perceived stress and social support among two different cohorts of Dutch medical students, and tested whether changes in stress and social support were related to the COVID-19 pandemic. This is an interesting sample, and addresses questions that are relevant to stress and coping during the pandemic. Although the paper is written in a very succinct style, many details of the study were clearly articulated. I’ll address my comments below in approximately the order they occur in the manuscript.

Introduction

-The introduction seemed unnecessarily minimal to me. The authors might want to increase their discussion of social support, especially since they go on to use/describe three different types of social support (perceived informational/emotional and the club membership variable). The club membership variable might best be understood as an indicator of engagement with social networks. This is important because the stress-buffering effects of social support are typically most clearly supported for measures of perceived social support (vs. social network engagement). It would be good to discuss why these measures were used and how they are relevant to medical students during the pandemic.

-In addition, the dependent measure was perceived stress, rather than a concept like resilience or other mental health outcomes. It would be useful to more clearly articulate why it is appropriate/useful to use naturally occurring COVID stress as an implicit IV that distinguishes the two cohorts, but also to look at perceived stress as the outcome. The authors hinted at a good reason when they indicated that social support may affect stress appraisal, but it would be better to more fully articulate/justify the approach taken by the study, especially since many of the studies they cite look at mental health outcomes.

-Overall, I found the writing to be clear, but there are some wording oddities that the authors should fix. For example, line 76 says “…social support can prevent that someone appraises…,” which is a very awkward construction.

Method

-I am confused about what is meant by line 122/123. Do the authors mean that they only got aggregate descriptive information on whole cohorts, and that variables like gender and performance are not linked with individual reports of PSS and social support? If so, I’m confused about how those factors were considered statistically. Could you please clarify?

-From the analysis, it looks like pre-and posttest data were paired for both cohorts. How were the responses paired (e.g., student ID number, etc?)?

-Were there missing data/items?

-Could you please say more about the shape of the distributions of PSS and social support? I am especially curious about the shape of the distribution of social support since the table notes that it is not normally distributed, but yet it seems to have been analyzed without modification in the regression analyses?

-I do not understand the function of the t-tests reported on lines 130-136. Why not just test these together as a 2 x 2 mixed ANOVA that considers time (fall-spring) and cohort (2018-2019)? It doesn’t make sense to separate the t-tests without first conducting the full 2x2.

-Did you consider looking separately at emotional and informational social support? It would be interesting to know whether the results were primarily driven by emotional social support, which likely would have been received from longer term relationships as opposed to new school-based relationships, since social connections were necessarily limited during the pandemic. That’s just speculation, of course, but it would be interesting to know if the results were similar.

-We are unable to see Table S-1, so unfortunately, I am unclear on the reliability/descriptives of these measures.

-As suggested above, it seems like a good idea to justify/contextualize the inclusion of the club measure as an indicator of social support.

Results

-The logic for the “post-hoc linear regression analysis” described on lines 184-189 was not at all clear to me. What is the purpose of this? Why cut off the continuous variable at the 25th percentile? Were you interested in the interaction between club membership and perceived social support? If so, why? Also if that were the goal, why not center both terms and make an interaction, as you did with the cohort x social support analysis (at least I assume you centered those predictors).

-There seems to be some sort of suppressor relationship going on with the interaction term. In Model 4 the b value for cohort jumps to 12.44 (from 3.89). Maybe see if this remains the same if the variables are centered (if they weren’t)? I also wonder whether skew in your variables (social support and PSS) might have contributed to this? Did you examine distributions and test for outliers? That might also help clarify the situation. It seems important to think about that change in the magnitude of the effect of cohort on stress in Model 3 (vs 2) and say something about it.

Discussion

-Thank you for highlighting the lower response rate in 2019. You might even take that a little further, since the response rate differences are confounded with the quasi-IV of pandemic stress. Were students equally likely to be missing both fall and spring reports in the 2019 cohort? If it data were especially likely to be missing in spring 2020, it would be possible to test whether those who dropped were different in PSS in the Fall (just through a t-test). Often it is the people who struggle the most who fail to continue in a longitudinal study. If that were the case, I think your argument would be that the selection differences underestimate the likely magnitude of the differences between the cohorts. If your most stressed people dropped from the study in Spring 2020, it would raise the mean for Spring 2020 in contrast to Spring 2019… and suggest that the actual difference in stress might be larger. See Cook and Campbell (1979) for a wonderful discussion of these selection effects, and possible selection x maturation effects.

Reviewer #2: The manuscript regarding university students and Covid was interesting and pertinent to the journal. The only issues were some of the tense used throughout was not correct (see below) with some punctuation issues. This reviewer commends the authors in their research.

Line 46, eliminate the dash after anxiety and depression.

Line, 53, eliminate the "-", maybe use a comma in place of them?

Line 55-57, use had instead of has, since the study is in the past. Same comment for the next two sentences, use was instead of is.

Line 64, change studies show to studies showed or use studies reported, but add the studies in this sentence.

Line 65, change review reports to reported.

Line 66, same comment as above, use reported.

Line 76, "Firstly, because..." not a complete sentence, please reword.

Line 77, "Secondly, because.."... is not a complete sentence, please reword.

Line 79, neighbours or neighbors?

Line 80, change shows to showed or reported.

Line 82, change indicates to indicated.

Line 83, change is to was

Line 92, check the punctuation, the dashes are not the same, maybe use commas?

Line 96, change is to was, since you did complete the student.

Line 106, make 6o, 60, use the actual zero, not the letter "o"

Line 133, change are to were.

Line 134, change is to was.

Line 141, change are to were

Line 142, change gender is to "gender was," change "performance is" to "performance was".

Line 150, change questionnaire is to "was".

Line 165, change are to was.

Line 166, change is to was.

Line 202, change is to was.

Line 221, confusing that "a" reference category is male, but above is female?

Line 232, change is to was.

Line 238, add a comma after 2018.

Line 241, change indicate to indicated.

Line 245, delete the dash, add a comma.

Line 269, change are to was.

Line 270, change do to did.

Line 271, change do to did.

Line 275, change demonstrates to demonstrated

Line 280, change indicate to indicated, is to was.

Line 302, change report to reported.

Lines, 323 and 324, delete the dashes and add commas.

6. PLOS authors have the option to publish the peer review history of their article (what does this mean?). If published, this will include your full peer review and any attached files.

Reviewer #1: No

Reviewer #2: No

---

## [Author Response · Author response to Decision Letter 0]

16 Jun 2022

See the attached file "Response to Reviewers"

---

## [Decision Letter · Decision Letter 1]

18 Aug 2022

PONE-D-21-39400R1Medical students’ crisis-induced stress and the association with social supportPLOS ONE

Dear Dr. Broks,

Thank you for submitting your manuscript to PLOS ONE. After careful consideration, we feel that it has merit but does not fully meet PLOS ONE’s publication criteria as it currently stands. Therefore, we invite you to submit a revised version of the manuscript that addresses the points raised during the review process.

Please find the reviewers' comments below this email and you will notice that one of the reviewers has still many concerns about the lack of clarity of using different tests/methods. Please address these concerns accordingly.

We look forward to receiving your revised manuscript.

Kind regards,

Muhammad A. Z. Mughal, PhD

Academic Editor

PLOS ONE

Reviewers' comments:

Reviewer's Responses to Questions

**Comments to the Author**

1. If the authors have adequately addressed your comments raised in a previous round of review and you feel that this manuscript is now acceptable for publication, you may indicate that here to bypass the “Comments to the Author” section, enter your conflict of interest statement in the “Confidential to Editor” section, and submit your "Accept" recommendation.

Reviewer #1: (No Response)

Reviewer #2: All comments have been addressed

2. Is the manuscript technically sound, and do the data support the conclusions?

Reviewer #1: Partly

Reviewer #2: Yes

3. Has the statistical analysis been performed appropriately and rigorously? 

Reviewer #1: No

Reviewer #2: Yes

4. Have the authors made all data underlying the findings in their manuscript fully available?

Reviewer #1: No

Reviewer #2: Yes

5. Is the manuscript presented in an intelligible fashion and written in standard English?

Reviewer #1: No

Reviewer #2: Yes

6. Review Comments to the Author

Reviewer #1: I have carefully reevaluated manuscript PONE-D-21-39400_R1, and appreciate the authors’ thorough attention to many of the issues raised in the previous reviews. I still think this manuscript is interesting and makes a good contribution to he literature. There are some remaining questions/concerns the authors and editor should probably consider as well. My most pressing concerns relate to the data analysis.

There are still some issues with the wording and use of tense throughout the manuscript. Perhaps a copy editor can assist with these details? For example, the sentence on line 33/34 in the abstract is unclear. Maybe instead say something like “Baseline stress measurements for the 2018 and 2019 cohorts did not differ significantly.” I also had to re-read the sentences on lines 97-98 and 185-188 to make sense of them. Line 242 should say memberS of a club… The use of “beside each other” in line 254/255 is awkward. Please review tense throughout—I noticed especially on p. 18 (lines 309-316).

I believe that clarity would be greatly improved if the authors used a term like “perceived social support” instead of "emotional-informational" support. If the authors want to keep the E-I support label they should explain the construct and use the term “emotional-informational” support in the introduction. Right now the first time the readers see that term is in the method section, and so it adds bulk and confusion throughout the last half of the paper. My sense is that the authors are using the MOS subscale as a measure of perceived social support, and so that label might work well. It looks like it is mainly an emotional social support measure. Only the last question appears to deal with informational support, and even that item seems to focus more on whether the person feels like they have someone available for informational support, rather than on the actual receipt of informational social support.

Likewise, although the discussion of social support is much improved in the introduction, there is still not a clear link to the club variable in the method section. School clubs could easily be added as an example of a setting that might be more likely to foster weak social ties (around line 82), so the reader was better prepared to see the club variable as a proxy social support variable.

Either the text or Table 1 should include standard deviations together with means.

I still did not find an analysis of attrition. It would be very useful to know if the pretest stress level differed between those people who remained in the study and those who dropped out. This would help empirically address the question of whether the people who completed the spring measure (in both cohorts) were more or less stressed than those who remained in the study. This could be done with a 2 x 2 between ANOVA (first IV would be whether the participants completed the spring assessment, second would be 2018/2019 cohort) with baseline stress as the DV. This would indicate whether those who dropped out were more/less stressed in the fall, and also whether that pattern differed by cohort. If for some reason that isn’t possible, you could instead use two between-subjects t-tests to test each cohort separately.

I appreciate the additional details about the analyses and tests of assumptions. That said, extreme outliers can still have a dramatic effect on regression results. Did the authors look at the extent to which those very low in SS may be driving the effects? I especially wonder about this because of the choice to look at the lower/upper quartile in the follow-up analyses. I find it odd not to specifically consider the possible effects of outliers, especially with a predictor with such strong negative skew. It is a little hard for me to understand the overall effects since no means/standard deviations were provided for the social support variable. I wonder if the slope of the relationship between social support and stress would remain stat sig for the 2019 cohort without the couple data points with very low social support. Something like a log transformation of the social support variable (reflected) would help determine whether the extreme scores were driving the results. Tabachnick and Fiddel’s Multivariate Stats book has a great chapter on the effects of distributions/outliers methods for handling them.

I am a bit confused about the concern re: homogeneity of covariance (for using a 2 x 2 ANOVA instead of separate t-tests (line 226). My understanding is that the assumption of sphericity/homogenity of covariance is only relevant when the within subjects variable has more than two levels. Could the authors provide a bit more detail or a reference on their logic here? Maybe I’m misunderstanding or forgetting something?

The post-hoc tests described in the text (considering both club membership and social support) would usually be conducted as a follow-up to an interaction. Did the authors test for a 3-way interaction between cohort, club membership, and social support? That is implicitly what is being compared through those quartile examinations, so it seems a bit strange to test this combination of variables without first testing for the 3-way interaction.

Please take care to use the words increase/decrease only when discussing the differences over time in the PSS variable. For example, line 249 discusses E-I social support increasing, but since that was only a cross-sectional measure, so describing correlations as indicating “increases” in social support is misleading. The authors are describing the fact that those who reported MORE social support showed lower stress (decreased stress). I’d also recommend removing decrease from line 322 to avoid any implication that that was a longitudinal comparison.

Please also be careful about using causal language about the role of COVID-19. For example, line 303 discusses the idea that the COVID-19 outbreak ELEVATED stress levels… but yet all we know is that changes in stress were greater in the 2019 cohort than in the 2018 cohort. Those cohorts include different people, less contact with faculty, and many other differences in addition to the pandemic. It would be hard to imagine that the pandemic wasn’t an important factor, but we can’t draw causal conclusions from this study. Also eliminate the term “COVID-19 induced stress” in line 308.

Reviewer #2: Well done article. These authors addressed all of the points that this reviewer had made, plus the additions of several paragraphs did make the manuscript stronger.

7. PLOS authors have the option to publish the peer review history of their article (what does this mean?). If published, this will include your full peer review and any attached files.

Reviewer #1: No

Reviewer #2: No

---

## [Author Response · Author response to Decision Letter 1]

21 Sep 2022

See the attached file "Response to Reviewers"

---

## [Decision Letter · Decision Letter 2]

21 Nov 2022

Medical students’ crisis-induced stress and the association with social support

PONE-D-21-39400R2

Dear Dr. Broks,

We’re pleased to inform you that your manuscript has been judged scientifically suitable for publication and will be formally accepted for publication once it meets all outstanding technical requirements.

Kind regards,

Muhammad A. Z. Mughal, PhD

Academic Editor

PLOS ONE

Additional Editor Comments (optional):

Reviewers' comments:

Reviewer's Responses to Questions

**Comments to the Author**

1. If the authors have adequately addressed your comments raised in a previous round of review and you feel that this manuscript is now acceptable for publication, you may indicate that here to bypass the “Comments to the Author” section, enter your conflict of interest statement in the “Confidential to Editor” section, and submit your "Accept" recommendation.

Reviewer #1: All comments have been addressed

2. Is the manuscript technically sound, and do the data support the conclusions?

Reviewer #1: Yes

3. Has the statistical analysis been performed appropriately and rigorously? 

Reviewer #1: Yes

4. Have the authors made all data underlying the findings in their manuscript fully available?

Reviewer #1: No

5. Is the manuscript presented in an intelligible fashion and written in standard English?

Reviewer #1: Yes

6. Review Comments to the Author

Reviewer #1: The authors have thoroughly responded to the concerns raised in previous reviews. I think this manuscript is much improved, and will make a solid contribution to the literature. I did spot a couple of language type-os, but I imagine those will be sorted out during publication.

7. PLOS authors have the option to publish the peer review history of their article (what does this mean?). If published, this will include your full peer review and any attached files.

Reviewer #1: No

---

## [Editor Report · Acceptance letter]

23 Nov 2022

PONE-D-21-39400R2 

Medical students’ crisis-induced stress and the association with social support 

Dear Dr. Broks:

I'm pleased to inform you that your manuscript has been deemed suitable for publication in PLOS ONE. Congratulations! Your manuscript is now with our production department. 

Kind regards, 

on behalf of

Dr. Muhammad A. Z. Mughal 

Academic Editor

PLOS ONE